# Attentional Deficits Following Preterm Birth: A Systematic Review and Meta-Analysis

**DOI:** 10.3390/brainsci15101115

**Published:** 2025-10-16

**Authors:** Kathrin Kollndorfer, Darlene Alicia Hörle, Florian Ph. S. Fischmeister

**Affiliations:** Developmental and Interventional Neuroimaging Lab (DINLAB), Department of Biomedical Imaging and Image-Guided Therapy, Medical University of Vienna, 1090 Vienna, Austria; kathrin.kollndorfer@meduniwien.ac.at (K.K.); darlene.hoerle@meduniwien.ac.at (D.A.H.)

**Keywords:** sustained attention, selective attention, preterm birth, cognitive development, meta-analysis

## Abstract

**Objective:** Preterm birth has been associated with an elevated risk of a broad range of neurodevelopmental impairments, including attentional deficits. This systematic review and meta-analysis aimed to synthesize the existing evidence on sustained and selective attention in school-aged children born preterm. **Methods:** Following PRISMA guidelines, a comprehensive literature search was conducted across PubMed, Ovid MEDLINE, EMBASE, and Web of Science. Eligible studies included assessments of sustained and/or selective attention in children aged 5–12 years born before 37 weeks of gestation. Data from 15 studies (sustained attention) and 12 studies (selective attention) were analyzed using random-effects meta-analyses. Additionally, subgroup analyses were performed based on gestational age. **Results:** Preterm-born children showed significantly poorer performance in sustained (Hedges’ g = −0.31, *p* < 0.001) and selective attention (Hedges’ g = −0.27, *p* < 0.001) compared to term-born controls. While sustained attention deficits were consistent across all gestational age subgroups, selective attention deficits were more pronounced in very early and extremely early preterm-born children. Moderate to late preterm-born children showed less impairment in selective attention tasks. **Conclusions:** Preterm birth is associated with measurable and persistent deficits in both sustained and selective attention, with greater vulnerability in children born before 32 weeks of gestation. These findings underscore the importance of implementing early monitoring and intervention strategies specifically designed to support attentional development in this high-risk population.

## 1. Introduction

Preterm birth is associated with a variety of short-term and long-term health risks, neonatal complications, as well as socioeconomic challenges. The World Health Organization (WHO) defines preterm birth as birth before the completion of 37 weeks of gestation [1]. Preterm birth is further divided into three groups based on gestational age (GA): (1) extremely preterm (<28 weeks); (2) very preterm (28 to <32 weeks); and (3) moderate or late preterm (32 to <37 weeks). Due to the medical advances of recent decades, the survival rates of children born preterm has increased drastically, especially within the group of babies born extremely preterm. As the fetal neural system develops rapidly in the third trimester [2], preterm birth is associated with increased risks of motor problems, such as cerebral palsy or less severe motor impairments [3,4], behavioral issues [5], and impaired cognitive function [6,7]. Even moderate and late premature-born children without developmental disorders show an increased risk of mild learning difficulties and suboptimal academic achievement [8]. Moreover, according to recent research, preterm and even early term birth are associated with an increased risk of attention deficit/hyperactivity disorder (ADHD) diagnoses [9].

Attention has been discussed and investigated extensively over the last decades, and there are numerous definitions across multiple scientific research fields. Although there is a common agreement that attention plays an important role in perception and information processing in the brain, attention is not a clearly defined concept. Due to this heterogeneity in definition and concepts of attention, research investigating the development of attention and attention problems is highly inconsistent.

One frequently investigated aspect of attention is sustained attention. It refers to the ability to maintain attention over time and focus on a goal-directed stimulus. The ability to concentrate over a longer period is one major predictor of children’s academic achievement [10]. Most research on sustained attention has been conducted in adults, with only a few studies investigating this concept in children. Betts et al. [11] found a rapid development of sustained attention performance during early school age. Impairments of sustained attention were detected in various neurological and developmental disorders, such as specific language impairment [12], autism spectrum disorders [13], or traumatic brain injury [14].

In contrast, selective attention allows an individual to focus on one particular stimulus while simultaneously suppressing other distracting information. This stimulus selection has been extensively investigated in the context of visual search [15,16]. Selective attention develops during childhood [17,18]. Preschool-aged children typically fail to filter out irrelevant information and show patterns of divided attention. Thus, children process more information than a specific task requires [19], and the processing of relevant information may therefore be less efficient [20]. Research in the field of impaired attention further suggests that children with attention deficit hyperactivity disorder (ADHD) show premature rather than impaired patterns of selective attention [21].

The primary objective of the present review and meta-analysis is to gain a deeper insight into the development of attention in preterm-born children. Since attention is a complex and multifaceted construct, covering a broad range of components, we reviewed published data on attention in preterm-born children. We performed two meta-analyses for sustained and selective attention, respectively.

## 2. Materials and Methods

This meta-analysis followed the Preferred Reporting Items for Systematic Reviews and Meta-analyses (PRISMA) checklist [22], consisting of the following steps: literature search, study identification, data extraction, quality control, and meta-analysis calculation. Appendix A presents the page numbers where relevant information can be found in this systematic review. No protocol was prepared for this systematic review and it was not registered.

### 2.1. Search

A systematic search of PubMed, Ovid MEDLINE, EMBASE, and Web of Science was performed with no language or article type restriction. The end of the search date in public databases was 2 February 2025. Additionally, references from the retrieved papers were examined to identify further relevant studies. The exact search algorithm can be found in Appendix A.

### 2.2. Selection Criteria

To be included in this meta-analysis, studies had to fulfill the following criteria: (i) investigate children between five and twelve years of age, born preterm (before 37 weeks of gestation) and term-born controls; (ii) report specific results from cognitive testing for sustained and/or selective attention; (iii) written in English language; (iv) published as original research in a peer-reviewed journal; and (v) published in the year 2000 or later. Reviews, meta-analyses, and registered reports were excluded. As intrauterine growth reduction is a significant factor influencing cognitive outcomes following preterm birth, we excluded studies focusing on preterm-born children with intrauterine growth reduction (IUGR).

### 2.3. Study Identification and Data Extraction

Two investigators independently screened titles and abstracts of all non-duplicated references and excluded irrelevant articles. A final list was agreed upon, and discrepancies were resolved by consensus between the two authors. In the second step, full-text versions of retrieved papers were evaluated for eligibility by two authors independently. Again, discrepancies between the two authors were discussed and resolved by consensus. In case of multiple publications of the same dataset, the first publication of the respective dataset was included in the meta-analysis.

Two authors independently extracted data; disagreements were discussed and solved by consensus. If necessary, authors were contacted for additional data. The following information was extracted from eligible studies: sample sizes for preterm-born children and term-born controls, as well as means and standard deviations of attention testing. In addition, we extracted information on mean birth weight and mean gestational age for the preterm group.

### 2.4. Quality Control and Risk of Bias of Included Studies

The methodological quality and possible risks of bias were evaluated using a modified version of the Newcastle-Ottawa Scale [23]. The modified quality assessment scale is presented in Appendix A. For all included studies, the quality of research was rated on three domains: selection of study groups, comparability of study groups, and outcome (see Appendix A). Two reviewers performed the quality assessment.

### 2.5. Statistical Analysis

All analyses were conducted using SPSS 29.0. Effect sizes were estimated using Hedges’ g to account for biases in studies with small sample sizes [24]. A random effects meta-analysis using the restricted maximum likelihood (REML) model was performed to estimate standardized mean differences (SMD) using Hedge’s g. These SMDs were pooled across all included studies. SMDs of 0.2 are interpreted as small effects, 0.5 as medium effects, and 0.8 as large effects [24].

The homogeneity of SMDs of included studies was examined using Cochran’s Q-test and the I^2^ statistics. The I^2^ statistic ranges from 0 to 100%, whereas values of 25% refer to small, 50% to medium, and 75% to large heterogeneity [25]. In addition, Egger’s regression test was performed to account for publication bias [26]. Two-tailed *p*-values of 0.05 were considered significant results.

As most included studies reported scores for sustained and selective attention, two separate meta-analyses were conducted for sustained and selective attention to rule out any data dependency. If studies reported more than one score for selective or sustained attention, data were pooled by calculating a mean performance score and a mean standard deviation. Subgroup analyses were performed for both meta-analyses for the gestational age of the preterm-born group (very early and early preterm-born children vs. moderate and late preterm-born children). To investigate the impact of the age at testing on attentional performance, a meta-regression on both attentional domains (sustained and selective attention) was performed.

## 3. Results

In total, 683 articles were retrieved from PubMed, 845 from EMBASE, 700 from Ovid, and 792 from Web of Science, resulting in a total of 3020 articles. After removing 1832 duplicates, abstracts of 1188 articles were screened. We then assessed 295 full-text articles and scanned them for eligibility. Of these full-text articles, 277 were excluded for various reasons, including failure to report results from attention testing, investigating children outside the predefined age range, and missing information on gestational age, among others. For more details on the selection process, see Figure 1. In total, 15 studies were eligible for inclusion in the sustained attention meta-analysis, investigating 3029 preterm-born children and 1126 term-born controls. For the selective attention meta-analysis, 12 studies were eligible for inclusion, investigating 4468 preterm-born children and 2723 term-born controls. The characteristics of eligible studies are summarized in Table 1. Quality control of included studies revealed a large variability, ranging from two points to seven (maximum score). Detailed results can be found in Appendix A.

### 3.1. Sustained Attention

Preterm-born children performed significantly poorer in sustained attention tasks compared to term-born controls (Hedges’ g = −0.31, *p* < 0.001, CI −0.39 to −0.22). Heterogeneity assessment revealed small and insignificant results (I^2^ = 8.5, Q =21.65, *p* = 0.086). There was no evidence of publication bias (Egger’s regression test; *p* = 0.998). Funnel plots are shown in Appendix A. The subgroup analysis showed similar results for very early/early and moderate/late preterm-born children (very early/early: Hedges’ g = −0.32, *p* < 0.001; moderate/late: Hedges’ g = −0.28, *p* < 0.001), for details see Figure 2. Meta-regression revealed no significant impact of age at testing on the results (t = −1.438, *p* = 0.174, see Appendix A).

### 3.2. Selective Attention

Comparable to sustained attention, preterm-born children performed significantly poorer in selective attention tasks compared to term-born controls (Hedges’ g = −0.27, *p* < 0.001, CI −0.40 to −0.15). However, in selective attention tasks, a moderate to large heterogeneity was observed (I^2^ = 64.2, Q = 30.96, *p* = 0.001). Egger’s regression test revealed no significant publication bias (*p* = 0.447). Funnel plots for selective attention can be found in Appendix A. The subgroup analysis revealed differences between very early/early and moderate/late preterm-born children (very early/early: Hedges’ g = −0.37, *p* < 0.001, CI −0.51 to −0.23; moderate/late: Hedges’ g = −0.11 *p* < 0.001, CI −0.17 to −0.05), for details see Figure 3. Meta-regression revealed no significant impact of age at testing on the results (t = 0.904, *p* = 0.387, see Appendix A).

## 4. Discussion

Preterm birth affects about one in every ten infants in developed countries. Despite substantial improvement in neonatal medical care and increasing survival rates of extremely preterm-born infants, premature birth is still the leading cause of neonatal morbidity and mortality [45,46]. Previous research has shown that even low-risk preterm-born children without severe neurological disorders, such as cerebral palsy or hemorrhagic bleeding, have a higher risk for developmental delay, language impairment, vision or hearing impairment, motor deficits, as well as social or behavioral problems [47]. As attention is a fundamental process that underpins many other cognitive functions, the present meta-analysis and review focus on two major aspects of attention: sustained and selective attention. Preterm-born children had poorer performance scores in both attentional domains. Interestingly, no impact of gestational age was observed in sustained attention. In contrast, in selective attention, performance scores were lower in very early and early preterm-born children (children born before 32 weeks of pregnancy).

### 4.1. Development of Attention

Attention development is a dynamic process that starts in infancy and continues until adolescence. Various aspects of attention develop at different stages of childhood and are coupled with the development of specific brain areas [48]. In infancy, sustained attention is brief and largely stimulus-driven. From the age of two, the ability to maintain attention for longer intervals improves continuously. However, young children are still easily distractible, and sustained attention is often influenced by the novelty of a stimulus [49]. During middle childhood until adolescence, sustained attention improves significantly. Children become more resistant to distraction and can focus on a stimulus for a longer time [50]. These improvements are associated with the maturation of prefrontal areas of the brain, the anterior cingulate cortex, and specific white matter tracts [48].

The development of selective attention starts in the first year of life. However, during infancy, selective attention is largely exogenous, as infants mostly orient to salient stimuli [49]. During preschool age, children begin gaining more voluntary control over attention and stimulus selection. Significant improvement of selective attention occurs during school age, with a higher involvement of the dorsolateral prefrontal cortex, the frontal eye fields, and the posterior parietal cortex [51].

A more recent approach investigating attention is the neurocognitive model of attention, which divides attention into three distinct but interacting networks: alerting, orienting, and executive control [52]. The alerting network is responsible for achieving and maintaining arousal and vigilance. Basic alerting functions are already present in infancy, but alertness becomes more stable and efficient during childhood [53]. The orienting network is responsible for directing attention to relevant sensory input. This system is functional in infancy, as infants can orient to novel stimuli. However, it becomes more efficient with age and experience during the first years of life [53]. The executive control network monitors conflicts, regulates thoughts and behavior, and is responsible for inhibitory control. This system develops more slowly and continues to improve throughout childhood and adolescence. It comes along with maturation processes and structural changes in the brain, such as synaptic pruning, myelination, and long-range functional connectivity between frontal and parietal areas [54,55].

### 4.2. Variability of Attention Performance in Preterm-Born Children

Although most studies have detected poorer cognitive performance in preterm-born children compared to their term-born peers, preterm-born children do not consistently show a pattern of neurological and cognitive impairment. The main reason for these differences is the enormous heterogeneity of preterm-born children. Besides obvious factors, such as gestational age at birth and birthweight, multifaceted endogenous and exogenous determinants may influence later cognitive outcomes.

Prenatal conditions have a considerable impact on development in later life. One of the most important factors for poorer cognitive development is the cause of preterm birth, such as maternal hypertensive disorders, or (gestational) diabetes, IUGR, and placental malformations [46,47,48,49,50,51,52,53,54,55,56]. Being small for gestational age has been shown to increase the risk of developmental delay and cognitive and motor impairment [39,57,58,59,60]. Besides genetic factors, IUGR may result from maternal health issues, behavioral habits, and maternal infections, leading to a mismatch between the placental nutrient supply and the fetus’s demand [61]. This may have a long-term impact on functional brain networks and cognitive development [59].

Preterm-born children are more vulnerable to adaptation problems due to their immaturity. Thus, neonatal complications can be found more frequently in this population, which may influence later development. Especially infants born before 32 weeks of gestation have a higher risk of suffering from Respiratory Distress Syndrome, Bronchopulmonary Dysplasia, necrotizing enterocolitis, (late-onset) sepsis, retinopathy of prematurity (ROP), or cerebral palsy [62].

Environmental factors may also influence the development of a premature infant. Previous research has shown that parental education, socioeconomic background [63,64], and even the parenting style in the first years of life [65] contribute to later cognitive development.

Finally, the way of assessing attention has a strong impact on the outcome. Previous research on the assessment of executive functions has shown that parental or teacher questionnaires and cognitive testing may provide different results [66]. Whereas the first evaluates a child’s behavior in everyday life, the latter shows the cognitive potential in a highly standardized situation [67]. Additionally, the present meta-analysis reveals that differences occur due to the assessment of different aspects of attention: While in sustained attention, all preterm-born children performed poorer compared to their term-born peers, in selective attention, very early and early preterm-born children had lower scores compared to moderate and late preterm-born children.

Especially in moderate and late preterm-born children, who constitute the vast majority of preterm-born children, results of previous research have revealed inconsistent results. One reason for these inconsistencies may be that attentional differences during infancy and childhood are subtle and hard to detect with the available test procedures. Recent research revealed that although differences between moderate to late preterm-born and term-born children are often small at the group level, preterm-born children are more likely to develop suboptimal attentional profiles [68].

### 4.3. Structural and Functional Brain Alterations

Altered brain structures can be detected using MRI techniques even at a very young age. At the structural brain level, a meta-analysis found decreased grey and white matter volumes in very preterm-born children compared to term-born peers [69]. In addition, white matter abnormalities were observed during school age, which may result in less efficient information processing [38,70] and may therefore account for the observed attentional deficits. Even in preterm-born children with low risk for neurodevelopmental deficits, a regional brain volume reduction in the cerebellum, the hippocampus, and the corpus callosum was observed [71]. Using Diffusor Tensor Imaging (DTI), a lower Fractional Anisotropy (FA) was observed in preterm-born children. Typically, FA increases with age, which is interpreted as a maturation process, whereas decreased FA following preterm birth has been used to infer subtle injuries in respective regions. Impaired FA was found specifically in fiber tracts that innervate cortical regions involved in interference control [72]. Structural alterations may be a result of disrupted neurodevelopment processes during the last weeks of pregnancy, such as synaptogenesis, myelination, and cortical folding [73,74].

A functional imaging study investigating selective attention showed differences in brain activation during the task. Although activation patterns in preterm-born children and term-born controls involved the same areas of the brain, the preterm-born group showed less activation in all involved brain areas, especially when the cognitive load was increased [75]. Using resting-state fMRI, very preterm-born children showed altered within- and between-network connectivity, although they had similar selective attention and inhibitory control abilities. These findings suggest that preterm-born children use compensatory strategies to overcome functional network alterations [76].

### 4.4. Clinical Relevance and Treatment Options

Altered development of attention has already been observed during infancy, characterized by slower reaction times, slower attention shifts, and poorer attention scores in neuropsychological testing [77,78,79]. However, attentional deficits may become more obvious during school age: Increasing cognitive demand at school may exceed previously successful compensatory mechanisms, resulting in poorer academic achievement, such as arithmetic performance or reading comprehension, learning difficulties in general, or the need for school assistance [8].

As suboptimal attention has a high prevalence in preterm-born children, various studies have focused on interventions for attentional performance. Research on neonatal nutrition has shown that protein-enriched formula (PDF) did not improve long-term neurodevelopmental outcomes at school age [80]. In contrast, an early postnatal mother-infant transaction program after preterm birth resulted in fewer behavioral problems at school age and better academic achievement [81].

Early stimulation of executive functions using a neuropsychological training program (PEFEN) in preterm-born children at preschool age revealed significant improvement of multiple functions, such as verbal comprehension, verbal fluency, working memory, visual and verbal memory, and attention [82]. A game-format training of executive functions (BrainGame Brian) provided inconsistent results. Although the training showed promising results in a pilot study, including improvements in executive function and attention in preterm-born children [83], a randomized controlled trial did not demonstrate an effect on attention, executive function in general, or academic achievement [84,85].

## 5. Limitations

Several limitations of this meta-analysis should be acknowledged. First, although we applied strict inclusion criteria and performed a comprehensive literature search, the number of eligible studies was limited, particularly for selective attention, which may reduce the generalizability of our findings. Second, the included studies varied considerably in their methodology, including different cognitive tasks, outcome measures, and definitions of attentional constructs. This variability may have contributed to the observed heterogeneity, especially in the selective attention analyses. Third, information on important confounding variables such as socioeconomic status, parental education, neonatal complications, and cause of preterm birth was not consistently reported across studies. Since these factors are known to influence cognitive outcomes, the inability to account for them represents a potential source of bias. Fourth, we excluded studies focusing on children with intrauterine growth restriction (IUGR) to reduce confounding; however, this approach may limit the ecological validity of our results, as IUGR frequently co-occurs with preterm birth in clinical practice. Fifth, publication bias cannot be fully ruled out, even though funnel plot inspection and Egger’s regression tests did not indicate significant asymmetry. Sixth, most studies provided cross-sectional rather than longitudinal data, which restricts inferences about developmental trajectories of attention in preterm-born children. Finally, our review was not preregistered, which may introduce reporting bias despite our adherence to PRISMA guidelines.

## 6. Conclusions and Future Research

This meta-analysis provides evidence that preterm birth is associated with significant deficits in both sustained and selective attention during childhood. Preterm-born children consistently performed worse than their term-born peers on sustained attention tasks. Importantly, these deficits were observed across all gestational age subgroups, indicating that reduced sustained attention is a general vulnerability among preterm populations, irrespective of the degree of prematurity.

In contrast, selective attention deficits were more heterogeneous and showed a gestational age effect. While very early and early preterm-born children exhibited moderate deficits in selective attention, moderate to late preterm-born children performed better in selective attention tasks. This finding suggests that selective attention may be more sensitive to the degree of prematurity and related neurodevelopmental immaturity.

Overall, the results highlight that attention is a consistently affected domain following preterm birth, with selective attention being more variable and linked to gestational age. Given the central role of attention in cognitive and academic development, these findings underscore the importance of early identification and targeted intervention strategies to support attentional functioning in children born preterm.

Future research may aim to clarify the underlying neurodevelopmental mechanisms of attentional deficits in preterm-born children by integrating behavioral assessments and longitudinal neuroimaging approaches. A more detailed characterization of attentional profiles and their developmental trajectories could aid in the development of personalized therapeutic approaches and the identification of critical periods for individualized intervention.

## Figures and Tables

**Figure 1 brainsci-15-01115-f001:**
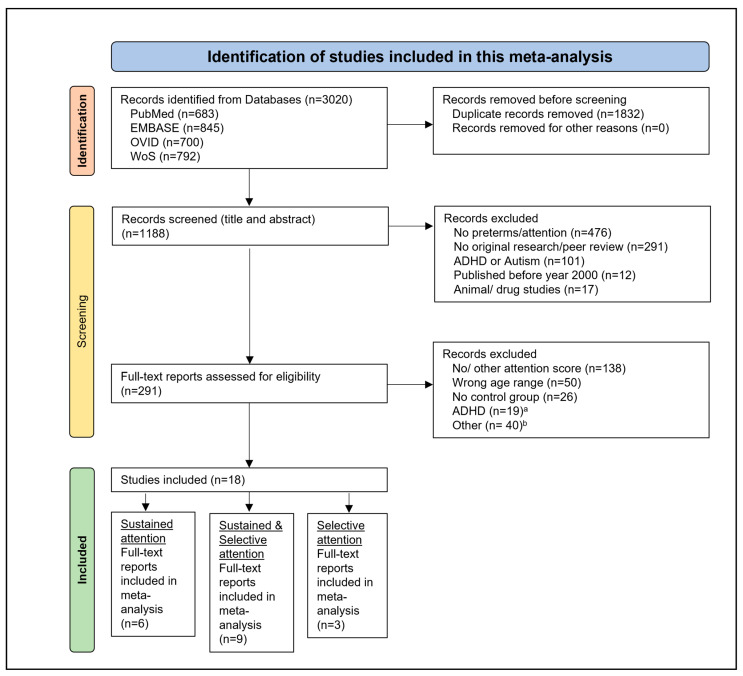
Study selection process following the PRISMA 2020 checklist [22]. ^a^ Studies focusing ADHD were excluded to give an overview of subclinical attentional performance. ^b^ Other reasons for exclusion (*n* = 40): Autism (*n* = 1), Case study (*n* = 1), Feasibility study (*n* = 1), Focus on birth weight, not gestational age (*n* = 10), intervention study (*n* = 4), multiple publication of an included sample (*n* = 9), Environmental health study (*n* = 1), Published before 2000 (*n* = 12), Attention score for only a part of the sample (*n* = 1).

**Figure 2 brainsci-15-01115-f002:**
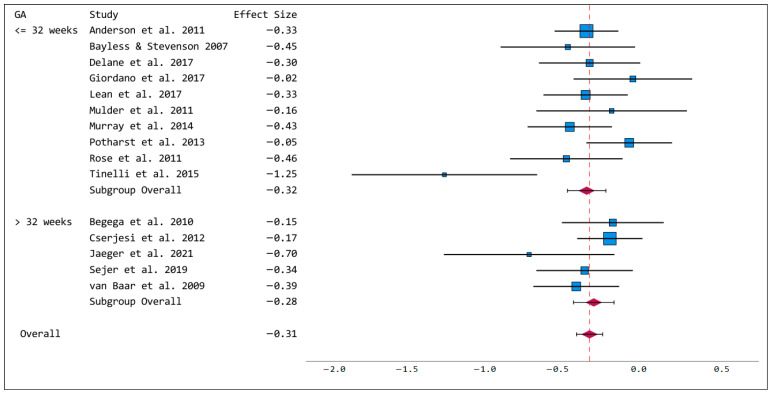
Forest plot of sustained attention with subgroups for gestational age ≤ 32 weeks and >32 weeks. Blue squares represent effect sizes of individual studies, red diamonds represent group effect size. References: Anderson et al., 2011 [27], Bayless & Stevenson 2007 [28], Delane et al., 2017 [31], Giordano et al., 2017 [32], Lean et al., 2017 [36], Mulder et al., 2011 [37], Murray et al., 2014 [38], Potharst et al., 2013 [39], Rose et al., 2011 [40], Tinelli et al., 2015 [43]; Begega et al., 2010 [29], Cserjesi et al., 2012 [30], Jaeger et al., 2021 [34], Sejer et al., 2019 [41], van Baar et al., 2009 [44].

**Figure 3 brainsci-15-01115-f003:**
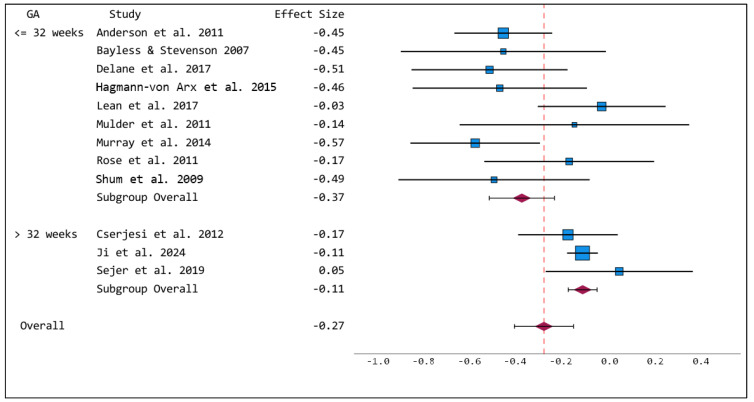
Forest plot of selective attention with subgroups for gestational age ≤ 32 weeks and >32 weeks. Blue squares represent effect sizes of individual studies, red diamonds represent group effect size. References: Anderson et al., 2011 [27], Bayless & Stevenson 2007 [28], Delane et al., 2017 [31], Hagmann-von Arx et al., 2015 [33], Lean et al., 2017 [36], Mulder et al., 2011 [37], Murray et al., 2014 [38], Rose et al., 2011 [40], Shum et al., 2009 [42]; Cserjesi et al., 2012 [30], Ji et al., 2024 [35], Sejer et al., 2019 [41].

**Table 1 brainsci-15-01115-t001:** Characteristics of eligible studies.

Study	Sample PT			Sample FT			Attention Task	Age at Testing
	N	Mean GA	Mean BW	N	Mean GA	Mean BW		
Anderson et al., 2011 [27]	189	26.5	833	173	39.3	3507	TEACh: Score! Sky Search	8 years
Bayless & Stevenson 2007 [28]	40	28.46	1200.75	41	-	-	TEACh: Score! Sky Search	8 years
Begega et al., 2010 [29]	63	33	2041	78	39–40	3102	d2 Selective Attention	8 years
Cserjesi et al., 2012 [30]	248	33.9	2239	130	39.7	3577	TEACh: Score! Map mission	6 years
Delane et al., 2017 [31]	77	27	940	66	40	-	TEACh: Score! Sky Search	7 years
Giordano et al., 2017 [32]	52	28.71	1172.91	52	39	3455.37	Sustained attention (KITAP)	5 years
Hagmann-von Arx et al., 2015 [33]	58	29.7	1302	55	39.7	3338	Selective attention (IDS)	8 years
Jaeger et al., 2021 [34]	31	32	1701	22	41	3662	Tonic alertness (modified from KITAP)	5 years
Ji et al., 2024 [35]	1706	32.1	2245	1865	-	3043	Flanker task	10 years
Lean et al., 2017 [36]	100	27.9	1063.5	106	39.5	3601.1	TEACh: Score! Sky Search	12 years
Mulder et al., 2011 [37]	56	27.6	-	22	-	-	TEACh: Score! Sky Search	9 years
Murray et al., 2014 [38]	198	27.4	960	70	39.1	3322	TEACh: Score! Sky Search	7 years
Potharst et al., 2013 [39]	102	28.71	1042	95	39.86	3436	Stop Signal Task	5 years
Rose et al., 2011 [40]	44	29.7	1165.2	86	-	-	CANTAB: Rapid Visual Information Processing Span of Apprehension	11 years
Sejer et al., 2019 [41]	40	35.8	2740.8	1728	40.29	3627.2	TEACh-5: Barking/Draw a line Great Balloon hunt/Hide and seek II	5 years
Shum et al., 2009 [42]	45	26.44	838.24	49	39.86	3577.84	NEPSY: Visual attention	8 years
Tinelli et al., 2015 [43]	29	28.3	1180	26	-	-	Visual sustained attention	8 years
van Baar et al., 2009 [44]	377	34.7	2425	182	39.5	3431	Bourdon-Vos Test	8 years

GA: gestational age (in weeks); BW: birthweight (in grams).

## Data Availability

The data supporting this study’s findings are available from the corresponding author upon request due to privacy and ethical restrictions.

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
