# Peer review of "Attentional Deficits Following Preterm Birth: A Systematic Review and Meta-Analysis"

_brainsci, 2025, doi:10.3390/brainsci15101115_

Round 1
Reviewer 1 Report
Comments and Suggestions for Authors
Thanks for this review.
Abstract: Perhaps reporting as 'very early' and 'extremely early' rather than 'early' would be preferential.
Methods: Please include search of grey literature and append the supplementary material and PRISMA flow diagram to include grey literature search.
Fig 1: n=19 studies ADHD were excluded, could the authors clarify why?
Fig 1: please clarify studies excluded for other reasons (n=40)
In the last part of the flow diagram: please rearrange as studies with selective attention only (n=..), studies with sustainable attention only (n=..) and studies with both (n=...), currently it is confusing as then total studies are 18, but then each type has 15 (missing 3) and 12 (missing 6 studies).
Please superscript 2 in I2 throughout the manuscript.
Please include the study quality as described in the Newcastle Ottawa Scale in the abstract and a summary in the main text.
Author Response
We appreciate the constructive and insightful comments from Reviewer 1, which helped us to improve the clarity and quality of our manuscript. In response, we have substantially revised our manuscript in accordance with the reviewer’s remarks.
1) Abstract: Perhaps reporting as 'very early' and 'extremely early' rather than 'early' would be preferential.
We thank the reviewer for this comment and refer to “very early” and “extremely early” in the revised version of the manuscript.
2) Methods: Please include search of grey literature and append the supplementary material and PRISMA flow diagram to include grey literature search.
We agree with the reviewer that grey literature is an important issue, as a lot of research may be missing without including these studies. However, we decided to explicitly exclude grey literature for two reasons:
- Gray literature is not peer-reviewed. Thus, there is no type of “quality control” from the scientific community.
- Gray literature submitted as preprints, conference proceedings or abstracts may represent preliminary analyses and, as such, often constitute a subset of late published manuscripts.
- For a part of the gray literature (mainly conference abstracts), a lot of information needed for the meta-analysis is missing, e.g. details on statistical testing, on the task used, or details describing the study population.
As stated in the methods section (Selection Criteria), we only included research published in peer-reviewed journals. In the revised version of the manuscript, we stated more clearly: “published as original research in a peer-reviewed journal”
3) Fig 1: n=19 studies ADHD were excluded, could the authors clarify why?
We decided to exclude studies on ADHD research because the majority of these studies do not use cognitive testing but symptom checklists. Moreover, participants with ADHD would show even poorer results in attention performance.
4) Fig 1: please clarify studies excluded for other reasons (n=40)
We thank the reviewer for this point. In the revised version of Figure 1, we explicitly list all other reasons in the figure caption.
5) In the last part of the flow diagram: please rearrange as studies with selective attention only (n=..), studies with sustainable attention only (n=..) and studies with both (n=...), currently it is confusing as then total studies are 18, but then each type has 15 (missing 3) and 12 (missing 6 studies).
In the revised version of Figure 1, we separated studies reporting sustained or selective attention only and those studies reporting both.
6) Please superscript 2 in I2 throughout the manuscript.
We corrected it in the revised version of the manuscript.
7) Please include the study quality as described in the Newcastle Ottawa Scale in the abstract and a summary in the main text.
We included the statement that study quality was assessed using a modified version of the Newcastle Ottawa Scale in the Abstract. In the main text, we added the range of the assessment. Detailed assessment is shown in the Supplementary Materials.
“Quality control of included studies revealed a large variability, ranging from two points to seven (maximum score). Detailed results can be found in Table S4.“

Reviewer 2 Report
Comments and Suggestions for Authors
General comments:
The objective of the systematic review is interesting. The authors have made an important exploration through SRMA on attention deficit associated with pre-term children. Though scientifically sound, there are other parameters that authors would need to consider strengthening their findings.
The authors have focused on collection of data from investigations conducted on the two attention parameters on preterm born children. In both cases the SRMA showed poorer attention in preterm children compared to controls, though no significant heterogenicity. Did the authors considered
- age of preterm born children at the time of investigation and its association with attention
- likewise, any other associated genetic or neurodevelopmental problem due to preterm and its association with attention
- gender difference if any
The result though explicitly mentions about the observations derived through meta-analysis; it is suggestive to extend the SRMA with association factors, that correlates with attention.
Discussion: This section is quite lengthy, and narrative. The authors have explained the various aspects of attention (development, structural and functional brain alterations) with respect to different levels of age, however no correlation is reported from their investigation and results through SRMA. I
Overall, the soundness of the objective and methodology of the study is scientifically interesting but lack some key observations. It is suggested to include the parameters for investigation that were informed in the discussion.
Author Response
We appreciate the constructive and insightful comments from Reviewer 2, which helped us to improve the clarity and quality of our manuscript. In response, we have substantially revised our manuscript in accordance with the reviewer’s remarks.
The objective of the systematic review is interesting. The authors have made an important exploration through SRMA on attention deficit associated with pre-term children. Though scientifically sound, there are other parameters that authors would need to consider strengthening their findings.
1) The authors have focused on collection of data from investigations conducted on the two attention parameters on preterm born children. In both cases the SRMA showed poorer attention in preterm children compared to controls, though no significant heterogenicity. Did the authors considered
- age of preterm born children at the time of investigation and its association with attention
- likewise, any other associated genetic or neurodevelopmental problem due to preterm and its association with attention
- gender difference if any
The result though explicitly mentions about the observations derived through meta-analysis; it is suggestive to extend the SRMA with association factors, that correlates with attention.
We fully agree with the reviewer that the above-mentioned parameters may influence the reported results. However, data on these parameters are rarely reported in the manuscripts. For instance, gender differences are reported explicitly in only one study (with non-significant results for selective attention and significant differences for sustained attention). Three more studies included gender in their regression model (not significant) or reported group x gender interactions (not significant). Although gender would be very interesting, the reported data do not allow for including this parameter in the meta-analysis.
The situation is pretty similar for genetic or neurodevelopmental issues. The majority of research excluded participants with severe disabilities (such as blindness, deafness or IQ <70). Some of the studies reported neonatal complications such as bronchopulmonary dysplasia, necrotising enterocolitis, or intraventricular haemorrhage. However, no data is reported on the impact of these conditions. Only one study included these medical conditions in the regression model, showing that bronchopulmonary dysplasia is a risk factor for poorer attention performance. However, due to the lack of information, we were not able to include medical conditions in the meta-analysis.
However, this point is added to the outlook as it is of special interest for future research.
“Large-scale studies investigating risk factors for poorer attention are highly needed, as many potential impact factors, such as potential gender differences or neurodevelopmental issues, are not investigated in detail yet.”
In the revised version of the manuscript, we include the analysis of age at testing, which provided no significant results (see main document results section (sustained and selective results respectively and Figure S3a and S3b).
2) Discussion: This section is quite lengthy, and narrative. The authors have explained the various aspects of attention (development, structural and functional brain alterations) with respect to different levels of age, however no correlation is reported from their investigation and results through SRMA. I
We tried to shorten the discussion a bit and rewrite it to make it more concise. However, we think that all of these subchapters are of interest. We added the information that age at testing was not a significant predictor of attention performance.
3) Overall, the soundness of the objective and methodology of the study is scientifically interesting but lack some key observations. It is suggested to include the parameters for investigation that were informed in the discussion.
We thank the reviewer for his/her constructive comments and tried to include further parameters. However, most of them were not reported in the original research. However, we analyzed the impact of age at testing on attentional performance using a meta-regression.
